# Deciphering how early life adiposity influences breast cancer risk using Mendelian randomization

Marina Vabistsevits [1,2 ✉], George Davey Smith [1,2], Eleanor Sanderson[1,2], Tom G. Richardson[1,2,3,5], Bethan Lloyd-Lewis[4,5] & Rebecca C. Richmond [1,2,5]

Studies suggest that adiposity in childhood may reduce the risk of breast cancer in later life. The biological mechanism underlying this effect is unclear but is likely to be independent of body size in adulthood. Using a Mendelian randomization framework, we investigate 18 hypothesised mediators of the protective effect of childhood adiposity on later-life breast cancer, including hormonal, reproductive, physical, and glycaemic traits. Our results indicate that, while most of the hypothesised mediators are affected by childhood adiposity, only IGF-1 (OR: 1.08 [1.03: 1.15]), testosterone (total/free/bioavailable ~ OR: 1.12 [1.05: 1.20]), age at menopause (OR: 1.05 [1.03: 1.07]), and age at menarche (OR: 0.92 [0.86: 0.99], direct effect) influence breast cancer risk. However, multivariable Mendelian randomization analysis shows that the protective effect of childhood body size remains unaffected when accounting for these traits (ORs: 0.59–0.67). This suggests that none of the investigated potential mediators strongly contribute to the protective effect of childhood adiposity on breast cancer risk individually. It is plausible, however, that several related traits could collectively mediate the effect when analysed together, and this work provides a compelling foundation for investigating other mediating pathways in future studies.

[1] Medical Research Council Integrative Epidemiology Unit at the University of Bristol, University of Bristol, Oakfield House, Oakfield Grove, Bristol, BS8 2BN, UK. [2] Population Health Sciences, Bristol Medical School, University of Bristol, Oakfield House, Oakfield Grove, Bristol, BS8 2BN, UK. [3] Novo Nordisk Research Centre, Headington, Oxford, OX3 7FZ, UK. [4] School of Cellular and Molecular Medicine, University of Bristol, Biomedical Sciences Building, Bristol BS8 1TD, UK. [5] These authors jointly supervised this work: Tom G. Richardson, Bethan Lloyd-Lewis, Rebecca C. Richmond. ✉email: marina.vabistsevits@bristol.ac.uk

Breast cancer is the leading cause of cancer-related deaths among women, with 1 in 8 at risk of developing the disease in high-income countries[1,2]. Although breast cancer mortality has declined over recent decades, mostly due to improved treatments and personalised diagnoses[3], the incidence of the disease has been steadily increasing by 3.1% annually since the 1980s[4]. Reducing the incidence rates will depend on better understanding and communicating modifiable risk factors both population-wide and targeted at women with an increased risk[5].

The second-largest preventable risk to all cancers after smoking is obesity[6], which has been extensively studied in relation to breast cancer. In observational studies, there is consistent evidence for a positive association of increased body mass index (BMI) with post-menopausal breast cancer, but an inverse association with pre-menopausal breast cancer[7,8]. This may be explained by varying levels of exposure to oestrogen in overweight compared with normal-weight women. Pre-menopause, overweight women have longer anovulatory cycles, thereby decreasing their exposure to ovarian hormones which has been suggested to reduce their breast cancer risk[9]. Post-menopause, adipose tissue is the main source of oestrogen biosynthesis[10]. This increases exposure to oestrogen in overweight compared to normal-weight women, which may explain their higher risk of breast cancer after menopause.

While previous conclusions have largely been drawn from observational epidemiological studies[11–13], several Mendelian randomization (MR) analyses have shown a contrary result, where genetically instrumented BMI is inversely associated with the risk of both pre- and post-menopausal breast cancer[14,15]. Since the genetic variants used to instrument BMI are set at birth, they should not be affected by environmental factors in later life. Thus, Guo et al.[14] hypothesised that the positive association between high BMI and post-menopausal breast cancer risk seen observationally may reflect adiposity and weight gain later in life. This is supported by findings of an inverse association between early-life (childhood and adolescence) BMI with breast cancer risk[11,12]. A recent MR study[16] using genetic variants related to childhood body size showed the same protective effect on breast cancer risk. Furthermore, using a multivariable MR analysis that accounted for adult body size, this study indicated that the protective effect of childhood body size influences breast cancer risk directly, independently of adult body size. However, the mechanism by which larger body size during childhood may reduce future breast cancer risk is not understood. Deciphering the mediating pathway between early-life adiposity and breast cancer would be of great interest for identifying targets of intervention, since advocating weight gain in childhood is not recommended.

Many established breast cancer risk factors are plausible candidate mediators of the protective effect of early-life body size, since they have been found to be influenced or associated with childhood adiposity. For example, body fatness during childhood may protect against breast cancer risk through hormonal pathways (e.g., IGF-1, oestradiol, testosterone, SHBG[17,18]). It has also been shown that incidence of breast cancer, particularly oestrogen receptor-positive (ER+) breast cancer, is substantially driven by changes in reproductive patterns, including parity and age at menarche, first birth and menopause[19,20]), events that are also influenced by increased adiposity in childhood[21,22]. Physical traits such as breast mammographic density (MD), which is an established risk factor of breast cancer[23], are also affected by body fatness in early life[24,25]. Finally, glycaemic traits have been extensively studied in relation to BMI and, while they have been inconsistently associated with breast cancer[26–28], are also of interest as potential intermediates.

In this work, we aimed to decipher the link between increased childhood body size and breast cancer risk by assessing a variety of potential mediators of this effect within a MR framework[29,30]. We characterised the effects of four groups of mediators that may be influenced by childhood body size to affect breast cancer risk: sex hormones, reproductive traits, glycaemic traits, and physical traits (Fig. 1). This was done using several extensions of the basic MR principle, including two-sample MR[31], two-step MR[32], and multivariable MR[33,34], with results then integrated into a mediation analysis.

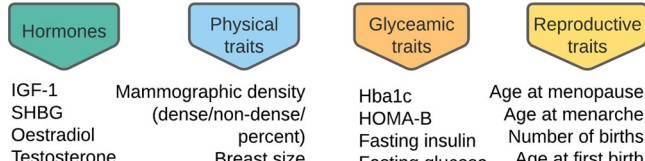

**Fig. 1 Potential mediator groups.** Traits that may be influenced by childhood adiposity to affect breast cancer.

## Results

**Study overview.** Using genome-wide association study (GWAS) summary statistics for childhood body size[16], breast cancer[35] and 18 hormonal, reproductive[36–38], glycaemic[39–44], and physical traits[15,45–47] (Fig. 1), we set up a workflow to investigate the role of these traits as plausible mediators of the childhood body size effect on breast cancer risk in later life (Fig. 2a). Firstly, we performed a two-step MR to identify traits influenced by childhood body size that have a causal effect on breast cancer. Subsequently, traits were assessed in a multivariable MR analysis to assess the magnitude of their direct effect on breast cancer risk, followed by a mediation analysis to quantify the indirect effect. The main exposure and outcome datasets used in the analysis comprised 246K female participants from UK Biobank (childhood body size) and 228K participants from Breast Cancer Association Consortium, BCAC, (breast cancer cases and controls), respectively. An overview of the GWAS datasets used as mediators is presented in Table 1. This study is reported as per the guidelines for strengthening the reporting of Mendelian randomization studies (STROBE-MR)[48].

**Two-step Mendelian randomization.** For each mediator, we conducted a two-step MR, where each step is an independent univariable two-sample MR analysis (Fig. 2b(ii)). In step 1, childhood body size was used as the exposure and a mediator trait as the outcome; in step 2, the mediator was used as the exposure and breast cancer as the outcome. This analysis provided insights into the presence of a causal effect on/from the mediators and was used as a mediator prioritisation step. The results obtained using the IVW (inverse-variance weighted) method are presented in Fig. 3.

Among the hormones investigated, evidence of an effect in both MR steps was observed for IGF-1 and free and bioavailable testosterone (Fig. 3). Increased childhood body size was associated with a reduction in IGF-1 (effect size per standard deviation, −0.24, 95% confidence interval: [−0.33: −0.15]), while IGF-1 had a positive effect on breast cancer risk (odds ratio per standard deviation, OR, 1.08 [1.03: 1.15]). Bioavailable and free testosterone estimates were similarly affected by childhood body size (0.09 [0.02: 0.16] and 0.12 [0.05: 0.18], respectively), and also had a similar positive effect on breast cancer risk (OR: 1.12 [1.05: 1.2] and OR: 1.14 [1.06: 1.22], respectively). Total testosterone had a positive effect on breast cancer (OR: 1.15 [1.06: 1.24], however, there was little evidence to suggest it was affected by childhood body size (−0.01 [−0.07: 0.04]). Oestradiol (−0.08 [−0.15: −0.02]) and SHBG (−0.19 [−0.29: −0.08]) were both inversely

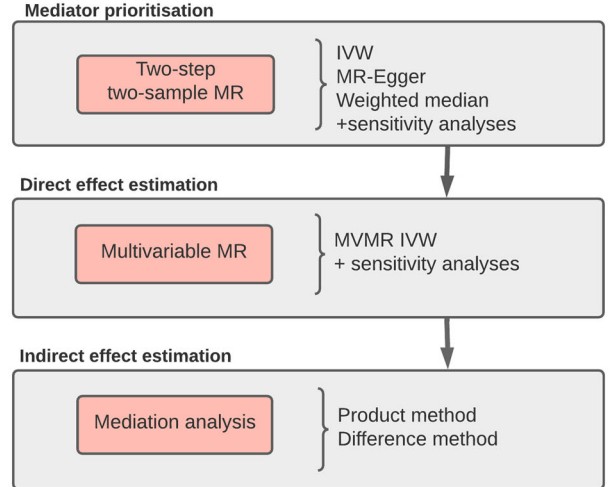

**a. Study design**

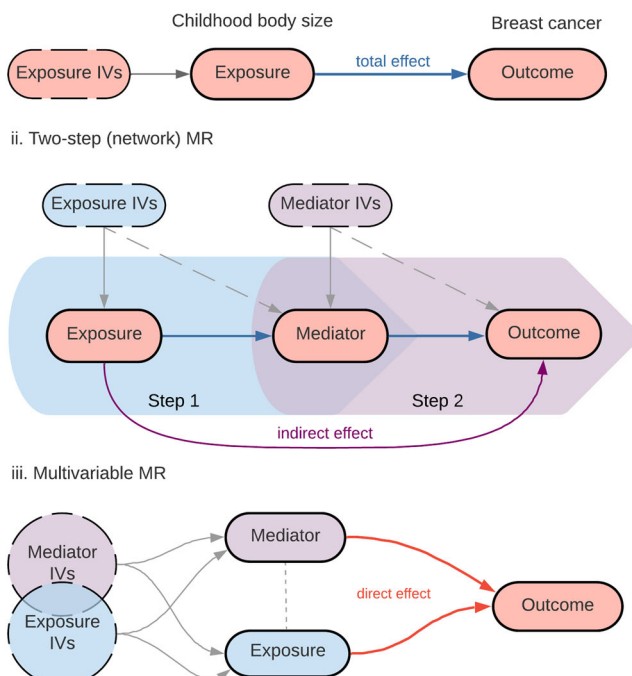

**b. Study methods**

**Fig. 2 Study design and Mendelian randomization (MR) methods used in the analysis. a** Study design; **b** Study methods: (i) Univariable MR: simple two-sample setup, measuring the total effect of exposure (childhood body size) on the outcome (breast cancer); (ii) Two-step (network) MR: two sets of two-sample MR — step 1: the total effect of exposure on the mediator, step 2: the total effect of the mediator on the outcome, allowing the measurement of the indirect effect of exposure on outcome via mediator; (iii) Multivariable MR: both exposure and mediator are accounted for in a single model; the direct effect of both is estimated. Arrow colours represent effect: blue — total, red — direct, purple — indirect. IVs — instrumental variables.

affected by increased childhood body size, but there was limited evidence of them affecting the risk of breast cancer (OR: 0.96 [0.68: 1.34] and OR: 0.97 [0.91: 1.04], respectively).

Analysis of reproductive traits showed the greatest effect of high childhood body size on age at menarche (−0.79 [−0.95: −0.64],

effect size per standard deviation, 95% CIs), as well as an effect on age at first birth (−0.09 [−0.16: −0.03]). There was little evidence that age at menopause and number of births were affected by increased childhood body size (0.02 [−0.34: 0.38] and −0.01 [−0.07: 0.05], respectively). Age at menopause was found to have a positive effect on breast cancer (OR: 1.05 [1.03: 1.07]). The OR point estimates of other reproductive traits were inverse, but overall, there was little evidence of their effect on breast cancer (age at menarche — OR: 0.98 [0.92: 1.05], age at first birth – OR: 0.92 [0.79: 1.07], number of births—OR: 0.70 [0.44: 1.11]). The estimates for age at menarche and age at menopause from the other data source were in agreement with the results in Fig. 3 (Supplementary Data 1 and 2).

The tested glycaemic traits generally had strong evidence of being positively affected by increased childhood body size, although there were some inconsistent results using different data sources for the same traits (e.g., fasting insulin, see Supplementary Data 1). The estimated effects were – fasting insulin: 0.16 [0.08: 0.24], fasting glucose: 0.05 [−0.01: 0.12], Hba1c: 0.07 [0.04: 0.11], HOMA-B: 0.1 [0.05: 0.15]. However, none of the glycaemic traits had a substantial effect on breast cancer risk (fasting insulin OR: 1.00 [0.58:1.72], fasting glucose OR: 1.03 [0.85: 1.25], Hba1c OR: 1.02 [0.74: 1.4], HOMA-B OR: 1.06 [0.78: 1.45].

For the physical traits, we were only able to perform the second step of the MR analysis (i.e., the effect of the trait on breast cancer) since we did not have access to the full GWAS summary data. There was limited evidence of an effect from breast size on breast cancer risk (OR: 1.11 [0.72: 1.71]), as previously shown by Nick Sern Ooi et al.[15] using the same data. Among the MD phenotypes, the dense area of the breast had a positive effect (OR: 1.39 [1.11: 1.73]) and the non-dense area had a negative effect (OR: 0.65 [0.46: 0.92]) on breast cancer risk. There was also a positive effect of percent MD, although with wider confidence intervals (OR: 1.43 [0.97: 2.12]).

We also repeated the second step MR analyses using breast cancer data stratified into oestrogen receptor-positive (ER+) and negative (ER−) groups[35] (Supplementary Data 3 and 4). Overall, similar findings were identified for the ER+ cases across all mediators, with stronger effects for MD phenotypes. None of the investigated hormones that had an effect on overall breast cancer risk showed evidence of an effect on ER− cases. Among the reproductive traits, the direction of effect switched from inverse to positive for age at menarche, whereas the estimate for age at menopause shifted closer to the null for ER− cases. No strong effects of the glycaemic traits were found on the ER− cases, consistent with ER+ and total breast cancer samples.

**Multivariable Mendelian randomization**. We next performed MVMR analyses with childhood body size and each mediator in turn, in relation to breast cancer risk (Fig. 2b(iii)). This allowed us to establish the direct effect of childhood body size on breast cancer risk after accounting for each mediator. MVMR was performed only on those mediators that showed the evidence of effect in at least one of the two-step MR steps and where the sensitivity analyses of both steps showed consistent results (see Sensitivity analysis and prioritisation logic in Supplementary Note 1, Supplementary Table 1).

The direct effects of childhood body size on breast cancer after accounting for each mediator estimated using the IVW-MVMR method are presented in Fig. 4 (Supplementary Data 5). Compared with the total effect of childhood body size on breast cancer from univariable MR, which had an OR of 0.66 [0.57, 0.76]) (Supplementary Data 6), the direct effects of childhood body size varied between OR of 0.59 (age at menarche) and 0.67 (total

**Table 1 Summary of GWAS datasets used as mediators in the study.**

| Mediator trait | Study | Sample size | Data source | GWAS size | Sample sex | SNPs at $5 \times 10^{-8}$ | Units | PMID |
|---|---|---|---|---|---|---|---|---|
| **Hormones** | | | | | | | | |
| IGF-1 | UK Biobank | 246,284 | IEU GWAS pipeline | 12.3M | F | 375 | SD | This study |
| Oestradiol | UK Biobank | 53,491 | IEU GWAS pipeline | 12.3M | F | 1 | SD | This study |
| SHBG | UK Biobank | 222,491 | IEU GWAS pipeline | 12.3M | F | 199 | SD | This study |
| Testosterone (bioavailable) | UK Biobank | 186,700 | IEU GWAS pipeline | 12.3M | F | 124 | SD | This study |
| Testosterone (free) | UK Biobank | 186,700 | IEU GWAS pipeline | 12.3M | F | 129 | SD | This study |
| Testosterone (total) | UK Biobank | 206,604 | IEU GWAS pipeline | 12.3M | F | 115 | SD | This study |
| **Reproductive traits** | | | | | | | | |
| Age at first live birth | UK Biobank[36] | 170,498 | OpenGWAS: ukb-b-12405 | 9.8M | F | 35 | SD | n/a |
| Age at menarche* | Perry (2014)[37] | 182,416 | OpenGWAS: ieu-a-1095 | 2.4M | F | 68 | SD | 25231870 |
| Age at menarche | UK Biobank[36] | 143,819 | OpenGWAS: ukb-b-3768 | 9.8M | F | 114 | SD | n/a |
| Age at menopause* | Day (2015)[38] | 69,360 | OpenGWAS: ieu-a-1004 | 2.4M | F | 42 | SD | 26414677 |
| Age at menopause | UK Biobank[36] | 243,944 | OpenGWAS: ukb-b-17422 | 9.8M | F | 200 | SD | n/a |
| Number of births | UK Biobank[36] | 250,782 | OpenGWAS: ukb-b-1209 | 9.8M | F | 11 | SD | n/a |
| **Glycaemic traits** | | | | | | | | |
| Fasting glucose* | Lagou (2021)[39] | 73,089 | MAGIC ftp | 8.7M | F | 22 | SD | 33402679 |
| Fasting glucose | Manning (2012)[40] | 58,074 | OpenGWAS: ieu-b-113 | 2.6M | M/F | 22 | SD | 22581228 |
| Fasting glucose | Scott (2012)[41] | 133,010 | OpenGWAS: ieu-b-114 | 64.5K | M/F | 35 | SD | 22885924 |
| Fasting insulin* | Lagou (2021)[39] | 50,404 | MAGIC ftp | 8.7M | F | 4 | SD | 33402679 |
| Fasting insulin | Manning (2012)[40] | 51,750 | OpenGWAS: ieu-b-115 | 2.6M | M/F | 4 | SD | 22581228 |
| Fasting insulin | Scott (2012)[41] | 108,557 | OpenGWAS: ieu-b-116 | 64.5K | M/F | 13 | SD | 22885924 |
| HBa1c* | Wheeler (2017)[42] | 123,665 | MAGIC ftp | 9M | M/F | 37 | SD | 28898252 |
| HBa1c | Soranzo (2010)[43] | 46,368 | OpenGWAS: ieu-b-103 | 2.5M | M/F | 11 | SD | 20858683 |
| HOMA-B | Dupuis (2011)[44] | 46,186 | OpenGWAS: ieu-b-117 | 2.4M | M/F | 4 | SD | 20081858 |
| **Physical traits** | | | | | | | | |
| Breast size* | 23andMe/Nick Sern Ooi (2019)[15] | 33,790 | supplementary material | 13M | F | 7 | Cup size | 31243447 |
| Breast size | 23andMe/Pickrell (2016)[86] | 33,790 | GWAS catalog/ supplementary material | 10M | F | 10 | Unit | 27182965 |
| Breast size | 23andMe/Eriksson (2012)[87] | 16,175 | GWAS catalog/ supplementary material | 10M | F | 6 | Cup size | 22747683 |
| Mammographic density* (dense/non-dense/per cent) | Sieh (2020)[47] | 24,192 | supplementary material | 650K | F | 15/10/9 | SD | 33037222 |
| Mammographic density (dense/non-dense/per cent) | Lindström (2014)[46] | -14,788 | GWAS catalog | 2.5M | F | 7/1/3 | Unit/% | 25342443 |
| Mammographic density (dense/non-dense/per cent) | Brand (2018)[45] | 9,498 | supplementary material | 500K | F | 5/2/3 | Volume (cm3)/% | 29665850 |

Mediator trait: name of the mediator as used throughout the paper; if there were several mediator GWAS available, * indicates the one that was used in the main analysis/plots. Study: cohort/publication/author of the dataset. Data source: how data were created or acquired; IEU GWAS pipeline is the MRC–IEU in-house GWAS pipeline for processing UK Biobank data[88], OpenGWAS[36] (gwas.mrcieu.ac.uk) is the MRC–IEU-based platform that hosts summary statistics of publicly available GWAS studies, the datasets IDs are provided; MAGiC ftp server: www.magicinvestigators.org/downloads/; 'supplementary material' refers to the data available with the publication in the 'Study' column; GWAS catalog[89] hosts data from GWAS studies. Sample size: number of samples used in the GWAS studies. Sample size: number of samples used in the GWAS. GWAS size: number of SNPs in GWAS summary stats. Sample sex: GWAS female/male composition. SNPs at $5 \times 10^{-8}$: number of genome-wide significant SNPs (top hits). Units: most are in standard deviation (SD) unless otherwise specified. PMID: PubMed ID.

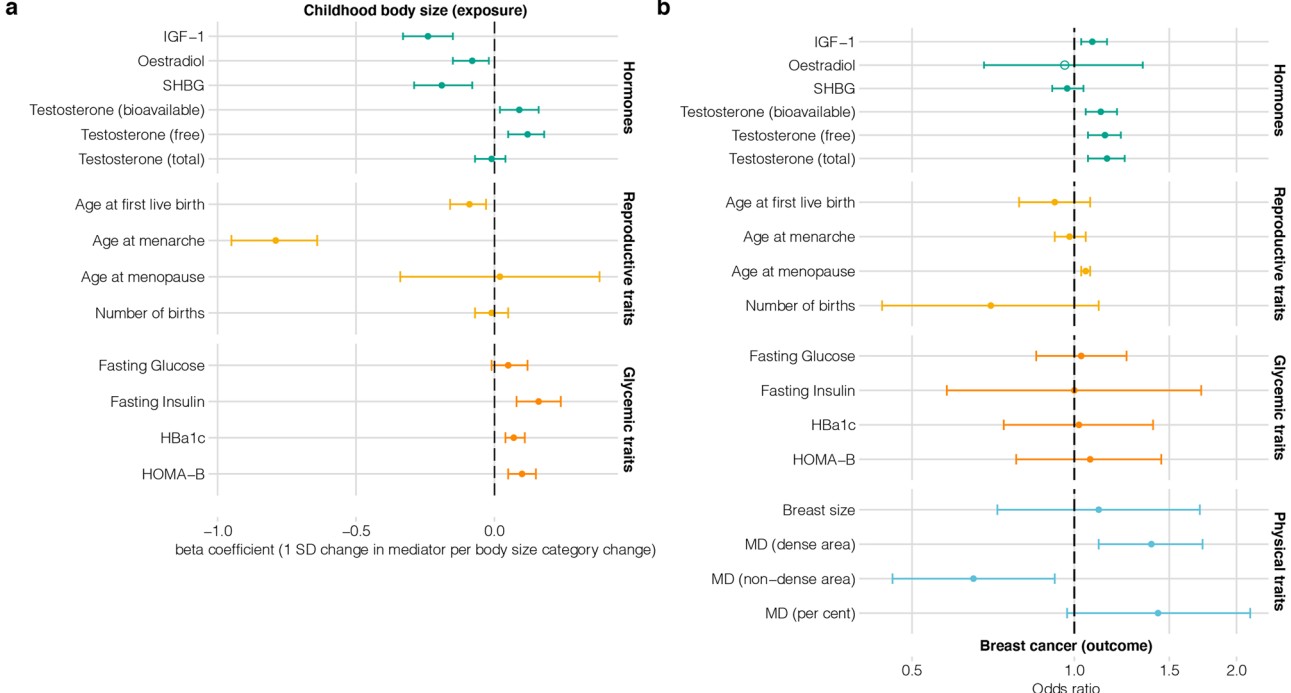

**Fig. 3 Two-step Mendelian randomization results evaluating four groups of potential mediators (hormones, reproductive traits, glycaemic traits, physical traits). a** Step 1: Plots showing the effect of childhood body size on the mediators (univariable MR). The effect is measured as the standard deviation (SD) change in mediator per body size category change. **b** Step 2: Plots showing the odds of breast cancer per SD higher mediators (unless otherwise specified in Table 1; univariable MR). Bars indicate 95% confidence intervals around the point estimates from IVW analyses (in step 1: effect size/beta, in step 2: odds ratio), except oestradiol (the estimate is based on a single Wald ratio and is indicated by an empty circle shape). MD — mammographic density. The presented data is available in Supplementary Data 1 and 2. The details about the datasets are provided in Table 1.

testosterone). This indicates no substantial change in the effect of childhood body size on breast cancer after accounting for each of the potential mediators, with effects consistent between overall, ER+ and ER− breast cancer (Supplementary Data 7 and 8).

We also performed analyses with adult body size and childhood height as third exposures in MVMR (in addition to childhood body size and the mediator) in two separate analyses. Again, no substantial change in the direct effect of childhood body size was observed in the presence of the additional exposures (Supplementary Data 9 and 10).

Overall, the direct effects of mediators on breast cancer risk were similar to their total effects once accounted for childhood body size (Supplementary Fig. 2, values in Supplementary Data 5). Among the hormones investigated, the direction and size of effect remained the same (±0.01 OR) for all measures (IGF-1, SHBG, and the three measures of testosterone). Among the reproductive traits, MVMR indicated a direct protective effect of increasing age at menarche once accounted for childhood body size (OR: 0.92 [0.86: 0.99]), and the effect was even stronger using the alternative age at menarche GWAS source (OR: 0.82 [0.74: 0.91], Supplementary Data 5). The positive effect of age at menopause remained consistent with univariable MR results (OR: 1.05 [1.03: 1.07]). The point estimates of Hba1c shifted from the null, but wide confidence intervals still indicate little evidence of the effect (OR: 1.06 [0.83:1.35]). We also estimated the direct effect on the ER+ and ER− breast cancer samples. The results for the ER+ cases were similar to those for the full breast cancer sample, but there was limited evidence of effect from mediators on the ER− sample, akin to the trend observed in step 2 of two-step MR.

In the MVMR analyses, additionally accounting for the adult body size effect, the mediator estimates were not considerably affected. However, accounting for childhood height reduced the

direct estimates for most mediators and attenuated IGF-1 and age at menarche effects to overlap the null (Supplementary Fig. 3 in Supplementary Note 3 and Supplementary Data 9 and 10).

**Sensitivity analysis.** To investigate the potential violations of the MR assumptions and validate the robustness of the two-sample MR results from the IVW approach, we performed additional MR analyses using MR-Egger[49] and weighted median[50] approaches, both of which are more robust to pleiotropy. The Egger intercept was used to explore the potential for the presence of directional horizontal pleiotropy, and Cochran's Q-statistic was calculated to quantify the extent of heterogeneity among SNPs, which is indicative of potential pleiotropy. The sensitivity analyses of MVMR included tests for instrument strength and horizontal pleiotropy. Performance in the sensitivity tests was used as a selection tool for mediator inclusion in the downstream analyses (Supplementary Note 1).

The estimated effects for hormones and reproductive traits were consistent across sensitivity analyses. The results obtained for some of the glycaemic traits, however, were variable and should be interpreted with caution. The sensitivity analysis details for all mediator groups are available in Supplementary Note 2 and Supplementary Data 11 and 12.

For all pairs of childhood body size and a mediator, which were prioritised for the MVMR analysis, conditional F-statistics were >10, indicating that weak instrument bias is unlikely to be present. The presence of directional pleiotropy was assessed by estimating $Q_A$ statistics, which consistently indicated excess heterogeneity and so the potential for pleiotropy. The MVMR sensitivity analysis results are presented in Supplementary Data 13–15.

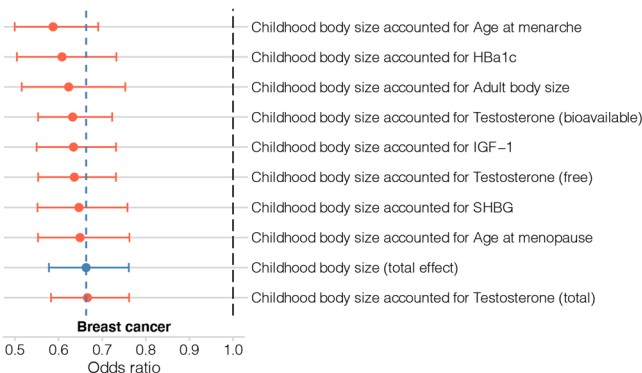

**Fig. 4 Multivariable Mendelian randomization results of childhood body size direct estimates accounted for selected mediators.** The plot shows the odds of breast cancer per SD change in the direct effect (red) of childhood body size accounted for mediators. The total effect (blue) (OR: 0.66 [0.57: 0.76]) and direct effect (i.e., accounted for adult body size, OR: 0.62 [0.51: 0.75]) estimates of childhood body size on breast cancer risk were re-calculated (Supplementary Data 6) from the original study [15] (using the thresholds defined in the Methods to match other MR analysis in this study), included here for comparison. The vertical blue line shows the value of the total childhood body size effect point estimate (0.66) relative to the direct estimates. Bars indicate 95% confidence intervals around the point estimates from IVW-MVMR. The estimated values and the numbers of genetic variants included in the analyses are given in Supplementary Data 5. The details about the datasets are given in Table 1.

Lastly, the violation of two-sample MR requirement of having two non-overlapping datasets for exposure and outcome traits, i.e., winner's curse, which was present for hormone traits (step 1 of two-step MR), was addressed with the split-sample approach (see Methods). Results of this analysis were consistent with the full sample analysis (Supplementary Note 4 and Supplementary Data 16 and 17).

**Mediation analysis**. Next, we carried out a mediation analysis to estimate the indirect effect of childhood body size on breast cancer risk via selected mediators, using the effect estimates from two-step MR, MVMR, and the total effect (Fig. 2b(i), Supplementary Data 6). This analysis was restricted to mediators that showed evidence of an effect in MVMR and that had substantial instrument strength (Supplementary Note 1).

Using a simulation analysis to compare available mediation methods (Supplementary Note 5), the Product method was chosen with reasonably high confidence as the main mediation analysis approach, with the Delta method as the corresponding SE/CI estimation technique. The indirect effect results are displayed in Fig. 5 and the estimates are available in Supplementary Data 18.

The indirect effects were modest, although there was some evidence for an inverse indirect effect via IGF-1 (−0.016 [−0.032: −0.003]), suggesting potential mediation via this trait. The estimated proportion of the mediated effect via IGF-1 was 0.039, indicating that any potential mediation via IGF-1 would only account for 3.9% of the total effect. Conversely, the alternative mediation approach (Difference method) estimated the IGF-1 indirect effect to be positive (described in Supplementary Note 5). Thus, the observed negative effect of IGF-1 requires further investigation. Mediation analysis also showed a positive indirect effect of childhood body size via age at menarche (0.065 [0.007: 0.126]) and via free and bioavailable testosterone (0.015 [0.005: 0.03] and 0.011 [0.002: 0.023] respectively), in contrast to the negative total effect of childhood body size on breast cancer.

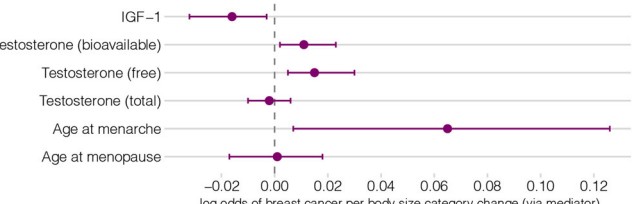

**Fig. 5 Mediation analysis results: indirect effect of childhood body size on breast cancer via a mediator.** The indirect effect was estimated as the product of coefficients of the total effect of exposure on the mediator (step 1 of two-step MR) and the direct effect of a mediator on the outcome (MVMR), i.e., Product method, and the 95% CIs were calculated based on SE estimated using Delta method. The presented data is available in Supplementary Data 18. The details about the datasets are provided in Table 1.

## Discussion

Previous observational and MR studies indicate that early life body size has a protective effect on the risk of breast cancer[11,12,16]. In this study, we investigated whether a number of breast cancer risk factors served as potential mediators of this protective effect using large genome-wide association datasets and a series of MR methods. Using two-step MR, we identified IGF-1, SHBG, testosterone, age at menarche and age at menopause as plausible mediators based on the effect of childhood body size on these traits, and their effect on breast cancer risk. However, when applied in a multivariable MR framework, none of these traits appeared to substantially mediate the protective effect of early life body size on breast cancer risk.

The results of our MR analysis of selected hormones on breast cancer are supported by recent MR studies, with similar effects observed for IGF-1 in Murphy et al.[51] and SHBG and testosterone in Ruth et al.[52] Conversely, the negative association of SHBG adjusted for BMI with breast cancer risk observed in Dimou et al.[53] had limited evidence of effect in our study, likely due to sex-specific analysis and differences in sample size. The effects observed for IGF-1 in two-step MR are in agreement with observational studies[17,54]. IGF-1 is known to play an important role in breast tissue differentiation and mammary gland function[55], and during normal development levels of IGF-1 gradually rise from birth to puberty, followed by a decrease in response to growth hormones[56]. The finding of lower adult IGF-1 in response to larger body size during childhood may be a part of the mechanism through which early life adiposity influences breast cancer risk. Our mediation analysis showed that childhood body size may have a negative effect on breast cancer indirectly via IGF-1. However, this result requires further investigation as the estimated indirect effect was relatively small (accounting for 3.9% of the total effect), and inconsistent across mediation methods. While there was some evidence for indirect effects of childhood body size via age at menarche, as previously reported[16,57], and testosterone levels, this was in the positive direction in opposition to the total inverse effect of childhood body size on breast cancer. Lastly, it was not feasible to confidently assess the effect of oestradiol on breast cancer in MR analyses due to the limited number of genetic instruments in the available data, but as oestradiol has been observationally associated with breast cancer[18], its mediating role remains of interest.

We also reviewed the effects in ER− and ER+ cancer samples, which can proxy for younger/older or pre-/post-menopausal women, respectively. The observed causal effects for hormones were maintained in the ER+ samples, but no effect was observed in the ER− samples. Some differences were also observed for reproductive traits, for example, age at first birth had a direct

effect only on the ER− cases (younger women). Additionally, there was a null effect of age at menopause and increased number of births among ER− breast cancer cases, which is reasonable since those exposures are less prevalent in younger women who typically present with ER− breast tumours.

Although we used the largest GWAS available for each mediator trait of interest, many of these data sources possess specific limitations that may have prevented us from identifying the mediated effect. When investigating hormones as outcomes, we used data from UK Biobank, which is the same sample as our main exposure (childhood body size). To assess whether sample overlap could have potentially led to winner's curse bias in step 1 of MR, a split-sample analysis was performed. Results of non-overlapping samples analysis were similar to the full sample analysis, suggesting that using the same data source for both exposure and outcome had little impact on our findings. Another limitation related to the hormones measures is that these were quantified in a predominantly post-menopausal sample of women (average age in UK Biobank is 56 years), where sex hormone levels are considerably different to those before menopause[58].

For the reproductive traits, we prioritised non-UK Biobank datasets in the main analysis to minimise the problem of sample overlap. While these datasets typically were from smaller sample sizes (and therefore fewer instruments), the directions of observed effects were consistent with the analyses performed on UK Biobank data (Supplementary Data 1).

The most inconclusive results were observed for glycaemic traits, likely due to smaller samples sizes and mixed-sex samples within these data sources. For traits with multiple available data sources, we prioritised those containing female-only participants[39], which typically reduced the sample size for the analyses but showed more relevant effects than in the mixed-sex analyses (Supplementary Data 1 and 2).

The unavailability of full summary data for the physical traits of interest is a major constraint in this study. Since only the top GWAS hits from MD studies by Brand et al.[45], Lindström et al.[46] and Sieh et al.[47] were available, we were unable to estimate the effect of childhood body size on these traits and the extent to which they could mediate the relationship with breast cancer. High MD is a major breast cancer risk factor and, importantly, is therapeutically modifiable[59]. Moreover, higher adiposity in childhood and adolescence has been associated with lower MD throughout adulthood[24]. In light of several recent studies[60,61] suggesting a plausible role of MD in the mediation of the protective effect of childhood body size on breast cancer risk, applying our MR framework to these datasets is an important aim for the future. We were also unable to perform the full analysis on breast size data. However, a previous study[15] using MVMR showed that breast size is unlikely to be a mediator of BMI effect on breast cancer risk.

While most of our analysis focused on mediators measured in adulthood, assessing mediators measured earlier in life would be useful for exploring the life-course effects of childhood body size. Investigating how childhood body size score influences plausible mediators over time, including an assessment of its effects on other anthropometric measures such as growth, changes in body composition and fat distribution[62], would provide another critical step to improve mechanistic understanding of its protective effect on breast cancer risk.

Childhood body size data from UK Biobank is based on a questionnaire completed by adult participants and could be subject to misclassification due to individuals misremembering their relative body size, which could potentially pose a great limitation to using this data. However, the genetic variants for childhood body size (originally identified by Richardson et al.[16])

were validated in several different cohorts[63,64] to be able to reliably separate childhood and adult body size, and also were robust to differential measurement error in simulations performed in the original study. Collectively, these analyses confirm the genetic variants from this data are suitable to be used to represent childhood body size.

Another important point to raise is the gene-environment equivalence assumption in MR, i.e., that if childhood body size is influenced genetically or environmentally this will have the same effect on the outcome[65]. It is necessary to consider whether childhood adiposity produced by environmental/lifestyle factors can reduce the risk of breast cancer in the same way as has been estimated using genetic variants that affect body size in early life.

It is also important to mention that current MR methods for mediation analysis assume linear associations. However, it is possible that the effects of childhood body size and mediators are non-linear, which could lead to an apparent lack of mediation despite the presence of the true meditating effect. Additionally, two-step MR and MVMR assume no interaction between the exposure and the mediator on the outcome. When assumptions of linearity and no interaction are not satisfied, the magnitude of the estimated effect may be affected[66].

In summary, here we systematically reviewed a set of potential mediators for the observed protective effect of increased childhood body size on breast cancer risk. Individually, none of the tested traits was found to strongly mediate this effect. However, it is plausible that several related traits may collectively contribute to the mediated effect, which could be explored in multi-mediator MVMR analyses in future studies. It would also be interesting to explore mediation effects on breast cancer experienced pre- and post-menopause, and ultimately by molecular subtypes of the disease. Mediation may also occur via a pathway that we have not considered in our study, or via MD, which could not be fully explored in this study due to the lack of full data availability. Finally, future work could explore proximal molecular mediators (e.g., breast tissue gene expression and methylation) to determine if early-life and adult adiposity have different effects on breast biology, which would be a critical step in deciphering the protective effect investigated in this work.

Our systematic investigation of mediators was designed as a prioritisation workflow, in which the initial MR analysis results (two-step MR) were used to select mediators for more advanced analyses (MVMR, mediation) based on the presence of causal effects and adequate performance in sensitivity analyses. Although we failed to identify a sole plausible mediator, we systematically report the MR results for the majority of obvious candidates from the largest available GWAS datasets to our knowledge. As new GWAS data becomes available, a similar approach can be applied, or used for investigating other biological questions (e.g., as shown in a recent study[67], aiming to identify the mediators of height effect on coronary artery disease). While we adopted a hypothesis-driven approach to investigate potential mediators, in future work, data mining platforms such as Epi-GraphDB (epigraphdb.org)[68] may be used to facilitate the identification of novel mediator traits/biomarkers, or candidates for multi-mediator MVMR analyses.

## Methods

**Data sources.** In this study, the main exposure trait GWAS (childhood body size) was from UK Biobank[69] and the outcome (breast cancer) from the Breast Cancer Association Consortium (BCAC)[35]. The sources of mediator traits GWAS are summarised in Table 1.

UK Biobank is a population-based health research resource consisting of approximately 500,000 people, aged between 40–69 years, who were recruited between the years 2006 and 2010 from across the UK. UK Biobank has received ethical approval from the UK National Health Service's National Research Ethics Service (ref 11/NW/0382) and informed consent from all participants. A full

description of the study design, participants and quality control (QC) methods have been described in detail previously[69]. The GWAS of childhood body size and adult body size used in this study were performed by Richardson et al.[16] on UK Biobank data ($N = 246,511$; female-only data, including for instruments extraction).

BCAC breast cancer GWAS includes 228,951 samples (122,977 cases and 105,974 controls) of European ancestry. The cases include both oestrogen receptor-positive ($N = 69,501$) and oestrogen receptor-negative ($N = 21,468$) participants[35]. The details of cohort design and genotyping protocol are described elsewhere (bcac.ccge.medschl.cam.ac.uk/bcac-groups/study-groups/, bcac.ccge.medschl.cam.ac.uk/bcacdata/). The study groups in the BCAC cohort do not include UK Biobank or mediator trait cohorts. The BCAC data was accessed through OpenGWAS[36] (gwas.mrcieu.ac.uk) under the following IDs: ieu-a-1126 (full sample), ieu-a-1127 (ER+), ieu-a-1128 (ER−).

**Description of selected traits.** Childhood body size is a categorical trait describing body size at age 10, with three categories ('thinner than average', 'about average', 'plumper than average'), from a questionnaire completed by adult participants of UK Biobank. Adult body size measure was converted from continuous adult BMI in UK biobank into three groups based on the proportions of childhood body size data to ensure that the GWAS results of both measures are comparable[16]. The genetic scores for childhood and adult body size were independently validated in two separate cohorts (HUNT Study (Norway)[63] and Young Finns Study[64]), which confirmed that the genetic instruments extracted by Richardson et al.[16] can reliably separate childhood and adult body size as distinct exposures.

Childhood height is another categorial trait from a questionnaire completed by adult UK Biobank participants, with three categories describing comparative height at age 10 ('shorter', 'about average', 'taller').

Four groups of mediators were assessed: sex hormones, female reproductive traits, glycaemic traits, and physical traits (Table 1).

(1) Hormones: IGF-1 (insulin-like growth factor 1), SHBG (sex hormone-binding globulin), oestradiol, testosterone (free, bioavailable, total). The different measures of testosterone were estimated as described in previous work[52]. The GWAS data for these traits was generated as a part of this study using the IEU GWAS pipeline (next section), and the consistency of genetic instruments was validated using LD Score Regression[70] with the published analyses of the same traits (Supplementary Note 6).

(2) Reproductive traits: age at first birth, age at menarche (×2), number of births, age at menopause (×2).

(3) Glycaemic traits: fasting insulin (×3), fasting glucose (×3), HbA1c (glycated haemoglobin A1c) (×2), HOMA-B (Homeostatic Model Assessment of β-cell function). HOMA-IR (insulin resistance) was considered in the analysis too, however, no robust instruments were identified.

(4) Physical traits: breast size (×3), MD (per cent, dense/non-dense area) (×3). Full summary data was not available for breast size and MD, so only GWAS top hits were used in the analysis.

Several traits were available from multiple data sources (marked with ×N). In the early stages of the analysis, all of them were evaluated, but only one version is presented in the final set of results. To prioritise a particular dataset over the rest we used the following criteria: (1) female-only sample, (2) non-UK Biobank data, (3) sample size. Table 1 shows the full set of tested datasets, highlighting the final selection with an asterisk.

**IEU GWAS pipeline.** The GWAS of hormone mediators from UK Biobank were performed using the MRC-IEU GWAS pipeline which is based on BOLT-LMM (v2.3)[71] linear mixed model and an additive genetic model adjusted for sex, genotyping array, and 10 genetic principal components. The data were inverse rank normalised prior to the analysis; the results are quantified as standard deviation change.

**Mendelian randomization.** MR is a type of instrumental variable (IV) analysis where genetic variants are used as proxies to uncover the causal relationship between a modifiable health exposure and a disease outcome[29]. There are three core assumptions that genetic variants need to satisfy to qualify as valid instruments for the causal inference: (1) variants have to be reliably associated with exposure of interest, (2) there cannot be any association with confounders affecting the exposure-outcome relationship, and (3) variants cannot be independently associated with the outcome, via pathway other than through the exposure (i.e., horizontal pleiotropy)[72].

The analyses in this work were performed using the two-sample (univariable) MR approach (Fig. 2b(i)), which relies on using GWAS summary statistics of two non-overlapping samples for exposure (sample 1) and outcome (sample 2)[73]. Two-sample MR is the basis for the more advanced analysis setup described in the next sections.

Two-sample MR analyses were performed using the inverse-variance weighted (IVW) method[74], which is presented in the Results. Alongside IVW, other complementary MR methods were applied to assess the robustness of the causal

estimates and to overcome any potential violations of the MR assumptions (e.g., horizontal pleiotropy) (see Sensitivity analysis for further details).

**Two-step MR.** Two-step MR (also known as network MR) is a sequence of two (or more) MR analyses connected by a shared variable. Two-step analysis setup is used to assess whether an intermediate trait acts as a causal mediator between the main exposure and the outcome of interest[32,75]. As shown in Fig. 2b(ii), in step 1, genetics variants (i.e., instrumental variables, IVs) for the exposure are used to estimate the causal effect of the exposure variable on the potential mediator, then, in step 2, the mediator IVs are used to assess the causal effect of the mediator on the outcome[75]. The evidence of a causal effect in both steps suggests that the association between exposure and outcome is mediated by the intermediate variable to some extent (further details in Mediation analysis).

**MVMR.** Multivariable Mendelian randomization (MVMR) is an extension to the standard univariate MR, which allows genetic variants to be associated with more than one exposure, and can estimate the direct causal effects of each exposure in a single analysis[33]. In this way, an exposure trait and a potential mediator can be analysed together to quantify the direct effect of both independently on the outcome (Fig. 2b(iii)). The genetic variants included in the analysis have to be reliably associated with one or both exposures but not completely overlap (i.e., no perfect collinearity), and have to satisfy the MVMR-extended second and third assumptions of the standard MR analysis[34,76]. Diagnostic methods and sensitivity tests for the robustness of MVMR estimates[76,77] are described under Sensitivity analysis.

**MR analysis tools.** All analyses were conducted using R (version 4.0.0). Univariate MR analyses and sensitivity tests were performed using the TwoSampleMR R package (version 0.5.4)[78], which was also used for accessing GWAS summary data deposited in OpenGWAS[36] (gwas.mrcieu.ac.uk). Multivariable MR was carried out by adapting TwoSampleMR's functionality to be used on mixed data sources (see "Code availability"). Sensitivity analyses for multivariable MR were performed using MVMR R package (version 0.2)[34].

For all exposure and mediator datasets, the instruments were extracted from the full summary data by selecting SNPs under $P$ value $< 5 \times 10^{-8}$ threshold and clumping the resulting set of variants with r2 < 0.001. When extracting instruments from the outcome (breast cancer) GWAS summary statistics, unavailable SNPs were substituted by proxies with a minimum linkage disequilibrium r2 = 0.8. The rest of the settings were kept to defaults as per package version number.

**Sensitivity analysis.** To further investigate the causal estimates found in the standard (IVW) MR analyses and to evaluate the validity of the analysed genetic instruments, MR-Egger[49] and weighted median MR[50] methods were used to overcome and accommodate for potential violations of the core MR assumptions. These complementary methods help to support the causal effects found with IVW, as a single method cannot account for all biological and statistical properties that may impact MR estimates. A variety of recommended[78] specialised sensitivity tests was also applied.

To assess overall horizontal pleiotropy (violation of assumption 3), the intercept in the MR-Egger regression[49] was evaluated, and the heterogeneity among the genetic variants was quantified using Cochran's Q-statistic[79]. The intercept term in MR-Egger regression is a useful indication of whether directional horizontal pleiotropy is driving the results of an MR analysis. When the Egger intercept is close to zero (e.g., < 0.002) and the P-value is large, this can be interpreted as no evidence of a substantial directional (horizontal) pleiotropic effect. When the Q-statistic for heterogeneity (difference in individual ratio estimates) is high and the corresponding p-value is small, this suggests evidence for heterogeneity and possibly horizontal pleiotropy. A high Q-statistic can be also used as an indicator of one or more variant outliers in the analysis, which may also be violating the MR assumptions.

Additionally, scatter plots of SNP effects from exposure and outcome fitted by all tested MR methods were evaluated for any deviations which would also be indicative of heterogeneity and violations of MR assumptions. Single SNP forest plots were used to summarise the effect of the exposure on the outcome due to each SNP separately, which is a helpful approach for visualising SNPs heterogeneity. Next, funnel plots were used to visually evaluate the direction of pleiotropy, which, if present, would be characterised by asymmetry in the plot. Finally, the sensitivity of causal inference to any individual genetic variant was tested by leave-one-out analysis, which is used to identify outliers.

In MVMR analyses, conditional F-statistics were used to evaluate the instrument strength[76], with F > 10 indicating suitable strength for the analysis. However, as in univariable MR, heterogeneity may be indicative of horizontal pleiotropy that does not act through one of the exposures. In MVMR, heterogeneity is quantified by $Q_A$-statistic (also a further modification of Cochran's Q), and small $Q_A$ indicates a lack of directional pleiotropic effect[76].

To calculate both statistic measures, a phenotypic correlation matrix for each MVMR test had to be constructed. This was done by applying the method for phenotypic matrix construction from GWAS summary data, available from metaCCA[80] R package. This is an alternative approach to using individual-level data for matrix construction. The method was only applied in cases when both

exposures in MVMR were from the same sample (i.e., UK Biobank). When data samples were different, the default settings for F and $Q_A$ statistics were used (i.e., gencov parameter set to zero)[76].

In the cases where MVMR sensitivity tests indicated the presence of weak instruments and potential pleiotropy via heterogeneity, Q-minimisation approach (Q-het) for estimating causal association was used to supplement the estimated of MVMR-IVW approach[76].

**Split-sample analysis**. One of the important requirements in two-sample MR is that the exposure and outcome GWAS are two non-overlapping datasets, which provides an advantage over the limitations in one-sample MR analysis of winner's curse and anti-conservative weak instrument bias[81,82]. In the two-sample MR analyses of childhood body size on hormones and some of the reproductive traits, this requirement was not satisfied, since the mediator GWASs were performed in the same cohort as the main exposure (childhood body size), i.e., UK Biobank.

To overcome the bias and evaluate the extent to which the results are affected, the MR analyses were repeated using a split-sample approach. This was possible due to the large sample size of the UK Biobank: the data were randomly split into two parts, and one part was used to carry out a new GWAS for the exposure (childhood body size), and the other part for the outcome (hormones/reproductive traits). The exposure instruments were extracted from the new split-sample exposure GWAS ($p < 5 \times 10^{-8}$), and step 1 of two-step MR and MVMR were repeated, with the resulting estimates are available in Supplementary Data 16 and 17 and Supplementary Note 4. Finally, the requirement of having two separate samples does not apply to exposures used in MVMR, i.e., it is acceptable to use UK Biobank traits for both exposure and mediator used in the analysis[34].

**Mediation analysis**. Mediation analysis is a method for quantifying the effects of an exposure on an outcome, which act directly, or indirectly via an intermediate variable (i.e., mediator)[66]. This analysis can identify which factors mediate the relationship between the exposure and the outcome enabling intervention on those mediators to mitigate or strengthen the effects of the exposure[83]. The total effect of exposure on outcome includes both a direct effect and any indirect effects via one or more mediators.

In terms of MR, the total effect is captured by a standard univariable MR analysis (Fig. 2b(i)). To decompose direct and indirect effects, we use the results from two-step MR and MVMR (Fig. 2b(ii) and 2b(iii)) in two mediation analysis methods: Difference method and Product method (Supplementary Note 5, Supplementary Fig. 5). For mediation analysis, it is important to have strong evidence of the total effect (Supplementary Data 6), effect in two-step MR and MVMR, and strong instruments in MVMR (measured by F-statistics) with no evidence of pleiotropy.

Lastly, although it is difficult to perform mediation analysis on binary outcomes (i.e., disease status) due to the non-collapsibility of odds ratios[84], it has been shown that if the outcome effects have been quantified as log-odds ratios, it is acceptable to use them for estimating the indirect effects[66]. However, it is important to note that the analysis on log-odds ratios from both MVMR and two-step MR is likely to have some bias for both rare and common binary outcomes[66]. When the outcome is common, like in the present study (53.7% cases), is it expected that the estimates from Product and Difference methods are not going to align, and both are likely to be biased, with Difference method producing a conservative estimate[85].

Unsurprisingly, we observed a disagreement between the estimates from the two methods in the results and therefore performed simulation analysis to help us choose the method that produces more reliable results in our study. The mediation results and simulation analysis are described in Supplementary Note 5. The mediation analysis approach that was selected via simulation was the Product method with Delta method for the indirect effect SE/CI estimation.

**Reporting summary**. Further information on research design is available in the Nature Research Reporting Summary linked to this article.

## Data availability
The GWAS datasets generated from UK Biobank and used in this study are available in OpenGWAS (gwas.mrcieu.ac.uk) and can be found using TwoSampleMR package filtering by PubMed ID or author (Rebecca Richmond). Source data underlying main figures are presented in Supplementary Data 20.

## Code availability
Mendelian randomization and mediation analysis code is available on GitHub (https://github.com/mvab/mendelian-randomization-breast-cancer) and Zenodo with https://doi.org/10.5281/zenodo.6349435. Simulation analysis for selecting the mediation method is available on GitHub (https://github.com/mvab/simulation_for_MR_mediation) and Zenodo with https://doi.org/10.5281/zenodo.6349442.

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

## Acknowledgements

B.L.L. is supported by the University of Bristol Vice-Chancellor's fellowship, Academy of Medical Sciences, Elizabeth Blackwell Institute for Health Research (University of Bristol) and the Wellcome Trust Institutional Strategic Support Fund; R.C.R. is a de Pass Vice Chancellor's research fellow at the University of Bristol; M.V. is supported by the University of Bristol Alumni Fund (Professor Sir Eric Thomas Scholarship). M.V., G.D.S., E.S., T.G.R., R.C.R. work in the Medical Research Council Integrative Epidemiology Unit at the University of Bristol supported by Medical Research Council (MC_UU_00011/1, MC_UU_00011/5 and MC_UU_00011/4). This work is also supported by a Cancer Research UK programme grant (the Integrative Cancer Epidemiology Programme) (C18281/A19169). This research has been conducted using the UK Biobank Resource under application number 15825.

## Author contributions

R.C.R., B.L.L., T.G.R., G.D.S. conceived and designed the study. M.V. cleaned the data, performed the analyses, interpreted the results, and wrote the initial draft of the manuscript as well as subsequent drafts with critical input on results interpretation from R.C.R., B.L.L., T.G.R., G.D.S., E.S. T.G.R., R.C.R., M.V. performed genome-wide association studies on UK Biobank traits. E.S. assisted with the simulation analysis and sensitivity analyses. The corresponding author attests that all listed authors meet authorship criteria and that no others meeting the criteria have been omitted.

## Competing interests

T.G.R. is employed part-time by Novo Nordisk outside of this work. All other authors declare no competing interests.
