## [Peer Review File · Communications Biology]

Reviewers' comments:

Reviewer #1 (Remarks to the Author):

Studies suggest that adiposity in childhood may reduce the risk of breast cancer in later life. Using a Mendelian randomization framework, we investigated 18 hypothesised mediators of the protective effect of childhood adiposity on later-life breast cancer, including hormonal, reproductive, physical, and glycaemic traits. presents a framework for the systematic exploration of potential biological mediators of disease in Mendelian randomization analysis.

This research is consistent with Mendelian randomization in research method and logic. However, there are problems in the following aspects: 1. The amount of data in blood glucose characteristics is too small, which reduces the intensity of analysis. 2. The data acquisition is not comprehensive enough and lacks the characteristics of childhood body shape data. 3. It is necessary to consider the impact of childhood obesity caused by the environment and lifestyle on the risk of breast cancer

Reviewer #2 (Remarks to the Author):

The authors conducted a very informative and careful analysis of the influence of childhood adiposity on breast-cancer risk later in life. The conclusion are fully supported by their work and discussion.

I would ask the authors to come up with a conclusion that actually reflects their (very relevant) contribution, not this "We have derived YET another framework". We may look back at this and say "this is a great piece of analysis, shall we use it as a template for our own work on X", and perhaps we should, but this is not a methodological paper at all (it does use a sound methodology though). All I am asking is to move some of this framework/approach label to a different place in the paper and simply focus on your contribution to answer the current biologically/clinically relevant question.

Some additional comments

1. Two what extend are the exposure and outcome (Breast cancer) GWAS completely independent? This is particularly important because weak-instrument bias may affect MVMR unknowingly through conditionally weak-instruments. The two sample setting would ensure this bias acts towards a null-effect. The authors know this full well I realize – just asking for clarification and if necessary discussion.

2. The authors go to some length to explain that a previous genetic score by Richardson for childhood obesity has been replicated. Despite this they independently select instrument (on sensible criterion), could the authors please explain why? Honestly, I am not necessarily a "fan" (if it matters) of transferable variants, so identifying instruments a new sounds defensible, just asking for clarification.

3. Related – it seems that childhood obesity was only available as self-reported variable with three categories.

a. How was the GWAS analysed, particularly with regards to the three categories – if some kind of proportional model was used, please describe how likely this proportionality assumption was. I don't particularly care if it did not hold perfectly, a reasonable approximation would be sufficient.

b. Could the authors please add a discussion item on the possible influence of differential measurement bias. For example, it seems reasonable to assume people who are thin in adult life may recall their childhood weight differently. How would this impact results?

4. Why was the MR-egger intercept test used to quantify potential horizontal pleiotropy bias. We already know it performs worse than the Q-statistic (which was also used). Clearly saying that the Q-statistic does not constitute horizontal pleiotropy bias when the intercept test is close to zero, this should account for the intercept tests variability. Hence you would be back to simply using the intercept test. This seems fairly circular in reasoning. Also should we start the discussion on "absence of evidence does not provide proof of absence". I understand the limitations of the Q-test, in this setting, but this does not seem to be doing much to address this, other than perhaps making the authors feel nice about themselves.

5. This statement

"Lastly, although it is sometimes advised against calculating an indirect effect if the outcome is a binary variable (i.e. disease status) due to non-collapsibility of odds ratios [60], it has been shown that if the outcome effects have been quantified as log-odds ratios, it is acceptable to use betas of such outcome in the mediation analysis [68]."

This is not necessarily false, but it does not very much reflect the literature and even the paper (despite the current authors being co-authors here). Clearly calculations of indirect effect will be biased when using odds ratios (or log odds ratio – do we really believe that taking a log addresses this at all?, the OR is primarily defined on the log scale, so it is only natural to use this scale for any manipulation). What the reference paper merely shows – through LIMITED simulations is that the bias may be minimal, especially when the disease is sufficiently rare (they arbitrarily say, as many would, that 10% is sufficient) . Instead of this somewhat cavalier and annoying statement, 1) acknowledge that these analyses are biased, 2) discuss that the bias may be minimal (refer to the incidence in the source GWAS please), especially due to other source of bias, 3) perhaps discuss the potential for effect direction flipping due to under/over-estimation related to non-collapsibility.

If need I will dig-up multiple reference that actually support this, but (I am already late with the review) reference 68 actually already support this and the authors merely use it in an attempted bluff.

At the same time, provided the above is discussed, I do appreciate the actual conducted work and do not mind a likely modicum of bias.

10. The figures are very nicely done, and appropriately scaled. Could the OR forestplot x-axis using a log-scale, with the tick labels exponentiated – this ensure the WALD-95%CI are symmetrical, preventing erroneous conclusions on a skewed in precision. Figure 4, step 1 – I presume the effect size has a unit, could this be added? Same for Figure 6 – beta is not a unit, it is a letter in the Greek alphabet.

Best wishes,

Florian Schmidt

Reviewer #3 (Remarks to the Author):

This is an interesting paper that adds to the literature on the effect of childhood adiposity on breast cancer, finding that effects of adiposity remain even when potential mediators are accounted for. The paper presents a framework for genetic epidemiology analyses that can also be used for addressing future similar research questions.

The paper is well written and presents comprehensive analyses and supporting data.

I suggest some specific comments for additional information to add, or points to clarify:

1. Was the childhood adiposity instrument based on the variants identified in females only?
2. Regarding GWAS generated in UK Biobank (e.g. sex hormones, IGF-1, number of births etc) that were used to create genetic instruments, what checks were done to ensure the validity of these signals and were they consistent with published analyses? For example, published GWAS of sex hormone measures in UKB have included various additional covariates (e.g. fasting time, menopause status).
3. Given that puberty timing is likely to be related to many of the other mediators investigated, e.g. IGF-1, hormone levels, have the authors considered exploring this in MVMR?

4. Supp Fig S3 and lines 231-232: Does attenuation of the effect estimates for age at menarche and IGF-1 with the addition of childhood height in the MVMR analyses suggest a more complex relationship between age at menarche and size in early life with breast cancer?
5. Lines 277-280, 304-305: How do you interpret the breast cancer increasing indirect effect of childhood size via age at menarche?
6. In the split sample analyses, did the instruments use variants identified at at $P < 5E-08$ from the full sample analysis or from the split sample analysis?
7. Line 227: I couldn't see 'age at first birth' in Table S11 or S12 and this trait isn't listed as prioritised in Supp Table S5.
8. Lines 231-232: I couldn't find the childhood height measures in the methods.

Response to reviewers

Please find below our detailed responses to all Reviewers' comments and suggestions marked in blue. *Italicised* text indicates text quoted from the manuscript. The line numbers mentioned in responses refer to the revised text. All changes in the revised text are tracked.

Reviewer #1:

This research is consistent with Mendelian randomization in research method and logic. However, there are problems in the following aspects:

⇒ Thank you for the positive appraisal of the manuscript and raising several valid queries. We have provided detailed clarifications for them below.

1. The amount of data in blood glucose characteristics is too small, which reduces the intensity of analysis.

Response:

We identified this as a potential issue during the study design, and therefore the analysis was performed in three separate studies of fasting glucose (all over 50K samples) for validation purposes. The estimates from the MR analyses were quite consistent across the studies:

Study	Sample sex	Sample size	nSNPs	MR step 1 (childhood body size on fasting glucose)	MR step 2 (fasting glucose on breast cancer)
Lagou 2021	F	73K	22	0.05 [-0.01: 0.12]	1.03 [0.85: 1.25]
Manning 2012	F/M	58K	22	0.05 [0.01: 0.09]	1.01 [0.82: 1.23]
Scott 2012	F/M	133K	35	0.10 [0.05: 0.15]	1.03 [0.84: 1.27]

We prioritised the Lagou 2021 dataset to be used in the main analysis as it is based on a female-only sample and has a reasonably large sample size (73K). However, it is the only dataset where the confidence intervals for the positive effect of childhood body size on fasting glucose crossed the null, although the effect estimate was consistent with that found in the mixed-sex samples.

We discuss this potential limitation of glycaemic traits on lines 402-405:

“The most inconclusive results were observed for glycaemic traits, likely due to smaller samples sizes and mixed-sex samples within these data sources. For traits with multiple available data sources, we prioritised those containing female-only participants [58], which typically reduced the sample size for the analyses but showed more relevant effects than in the mixed-sex analyses (Supplementary Tables S2 and S3).”

2. The data acquisition is not comprehensive enough and lacks the characteristics of childhood body shape data.

Response:

Childhood body size data acquisition is described in Methods on lines 551-552: *“Childhood body size is a categorical trait describing body size at age 10, with three categories (‘thinner than average’, ‘about average’, ‘plumper than average’), from a questionnaire completed by adult participants of UK Biobank.”*

The genetic score for this measure has been validated in several cohorts. In the original publication using childhood body size data [1], Richardson *et al* performed linkage disequilibrium score (LDSC) regression to validate that the UK Biobank measure of childhood body size is more strongly correlated with measured obesity in childhood ($rg = 0.85$) than in adulthood ($rg = 0.64$) (Richardson *et al* 2020, Supplementary Figure 3), using childhood BMI data from the EGG consortium [2] and adult BMI from the GIANT cohort [3].

Later, the genetic scores for childhood and adult body size were independently validated in two separate cohorts (HUNT Study (Norway) [4] and Young Finns Study [5]), which confirmed that the genetic instruments extracted by Richardson *et al* can reliably separate childhood and adult body size as distinct exposures.

Finally, LDSC regression analysis has been repeated with the most recent release of EGG consortium data of childhood BMI measure and showed a genetic correlation of $r_g = 0.96$ with UK Biobank childhood body size (unpublished results by Tom Richardson), suggesting that the UK Biobank data in this study provides a good representation of the genetic effects on childhood body size.

[1] T. G. Richardson, et al, 'Use of genetic variation to separate the effects of early and later life adiposity on disease risk: Mendelian randomisation study', *The BMJ*, vol. 369, 2020

[2] Bradfield JP et al, Early Growth Genetics Consortium. A genome-wide association meta-analysis identifies new childhood obesity loci. *Nat Genet* 2012;44:526-31.

[3] Speliotes EK, et al, MAGIC, Procardis Consortium. Association analyses of 249,796 individuals reveal 18 new loci associated with body mass index. *Nat Genet* 2010;42:937-48.

[4] M. Brandkvist et al., 'Separating the genetics of childhood and adult obesity: a validation study of genetic scores for body mass index in adolescence and adulthood in the HUNT Study', *Human Molecular Genetics*, vol. 29, no. 24, pp. 3966–3973, Dec. 2020,

[5] T. G. Richardson et al., 'Evaluating the direct effects of childhood adiposity on adult systemic metabolism: a multivariable Mendelian randomization analysis', *International Journal of Epidemiology*, Mar. 2021

3. It is necessary to consider the impact of childhood obesity caused by the environment and lifestyle on the risk of breast cancer

Response:

Thank you for bringing up this important point. We have included a discussion of the gene-environment equivalence assumption in the manuscript on lines 434-438. This assumption is key to interpreting MR results: perturbations caused by genotype have the same downstream effects as if they were caused by modifiable (environmental) exposures.

“Another important point to raise is the gene-environment equivalence assumption in MR, i.e. that if childhood body size is influenced genetically or environmentally this will have the same effect on the outcome [65]. It is necessary to consider whether childhood adiposity produced by environmental/lifestyle factors can reduce the risk of breast cancer in the same way as has been estimated using genetic variants that affect body size in early life.”

Reviewer #2:

The authors conducted a very informative and careful analysis of the influence of childhood adiposity on breast-cancer risk later in life. The conclusions are fully supported by their work and discussion.

I would ask the authors to come up with a conclusion that actually reflects their (very relevant) contribution, not this “We have derived YET another framework”. We may look back at this and say “this is a great piece of analysis, shall we use it as a template for our own work on X”, and perhaps we should, but this is not a methodological paper at all (it does use a sound methodology though). All I am asking is to move some of this framework/approach babel to a different place in the paper and simply focus on your contribution to answer the current biologically/clinically relevant question.

⇒ Thank you for the positive appraisal of the manuscript, your insightful comments have been extremely valuable. We provide responses and clarifications to your queries below.

Response:

We appreciate the point and have rephrased the mentions of ‘framework’ in the abstract (lines 44-46) and on lines 120 and 129.

Some additional comments

1. Two what extend are the exposure and outcome (Breast cancer) GWAS completely independent? This is particularly important because weak-instrument bias may affect MVMR unknowingly through conditionally weak-instruments. The two sample setting would ensure this bias acts towards a null-effect. The authors know this full well I realize – just asking for clarification and if necessary discussion.

Response:

The UK biobank is not a part of the BCAC breast cancer consortium (<https://bcac.ccge.medschl.cam.ac.uk/bcac-groups/study-groups/>). If there is any overlap in samples, it will be minimal and likely has a negligible effect on the overall results. The same goes for other smaller exposure cohorts used in the analysis.

The instrument strength was tested in all MVMR analyses and the results were considered satisfactory (Fst > 10) (Supplementary Table 8, also presented below).

mediator	Fst childhood body size	Fst mediator	QA-statistic	Qpval
IGF	21.2	70.1	740.7	1.06E-45
SHBG	26.8	75.6	803.2	6.04E-66
Testosterone (bioavailable)	36.3	44.8	445.8	1.33E-26
Testosterone (free)	35.4	45.0	494.3	1.82E-32
Testosterone (total)	36.7	43.9	471.8	2.64E-29
Age at menarche (Perry)	30.0	30.1	315.1	1.13E-21
Age at menopause (Day)	28.8	52.7	312.0	2.09E-24
HBa1c (Wheeler)	50.0	34.1	248.2	2.13E-20

2. The authors go to some length to explain that a previous genetic score by Richardson for childhood obesity has been replicated. Despite this they independently select instrument (on sensible criterion), could the authors please explain why? Honestly, I am not necessarily a “fan” (if it matters) of transferable variants, so identifying instruments anew sounds defensible, just asking for clarification.

Response:

The instruments in Richardson’s paper had been manually extracted, while in this project we used TwoSampleMR R package functions for instrument extraction and clumping to make sure that the

analysis was up to date with the latest protocols. In both cases, the same 1000GP reference was used, so any variation should be minimal. MR results obtained using the two approaches were very consistent:

	Richardson et al 2020	This study
Univariable MR (IVW)		
Childhood body size -> BC	0.63 [0.55: 0.72]	0.66 [0.58: 0.76]
Adult body size -> BC	0.82 [0.73: 0.92]	0.82 [0.72: 0.93]
Multivariable MR		
Childhood body size -> BC	0.59 [0.50: 0.71]	0.62 [0.52: 0.75]
Adult body size -> BC	1.08 [0.93: 1.27]	1.08 [0.91: 1.29]

3. Related – it seems that childhood obesity was only available as self-reported variable with three categories.

a. How was the GWAS analysed, particularly with regards to the three categories – if some kind of proportional model was used, please describe how likely this proportionality assumption was. I don't particularly care if it did not hold perfectly, a reasonable approximation would be sufficient.

b. Could the authors please add a discussion item on the possible influence of differential measurement bias. For example, it seems reasonable to assume people who are thin in adult life may recall their childhood weight differently. How would this impact results?

Response:

Childhood body size data acquisition is described in Methods on lines 543-545: “*Childhood body size is a categorical trait describing body size at age 10, with three categories ('thinner than average', 'about average', 'plumper than average'), from a questionnaire completed by adult participants of UK Biobank.*” For the purposes of the GWAS, the variable was treated continuously, so assuming proportionality between the categories.

In the original publication [1], LDSC regression analysis showed childhood body size from UK Biobank is more strongly correlated with measured obesity in childhood [2] ($r_g = 0.85$) than in adulthood [3] ($r_g = 0.64$) (Supplementary Figure 3 in [1]). Later, the genetic scores for childhood and adult body size were independently validated in two separate cohorts (HUNT Study (Norway) [4] and Young Finns Study [5]), which confirmed that the genetic instruments extracted by Richardson *et al* can reliably separate childhood and adult body size as distinct exposures. Finally, LDSC regression analysis has been repeated with the most recent release of EGG consortium data of childhood obesity measure and showed a genetic correlation of $r_g = 0.96$ with UK Biobank childhood body size (unpublished results by Tom Richardson), suggesting that the UK Biobank data in this study is suitable to use to represent childhood body size.

Since EGG and other studies have used continuous body size measures, the strong r_g value supports the proportionality assumption of the childhood body size measure. Moreover, a recent study [6] has shown that these childhood body size instruments explain more of the variation in measured BMI across childhood compared to instruments from the largest childhood BMI GWAS to date from the EGG consortium.

The issue of differential measurement bias was also explored and discussed in the original publication by Richardson *et al* in the analyses presented in Supplementary Note 2 [1]. They conducted a simulation study to identify what effect measure misclassification due to recall bias may have on the results. They modelled the data to have a true continuous effect on the outcome but classified in the categories to (0/1/2) to reflect the real body size data setup. Some of the scenarios that were investigated with the model were (1) individuals misremembering their childhood size as being

average like their adult body size, and (2) remembering their childhood size as being lower than it was. The results showed that there is some bias in the estimated causal effects due to misclassification, in the direction away from the null. Misclassification of childhood size to a random or lower category weakens the instrument strength of this variable in multivariable analysis with adult body size, and may bias adult effects when measurement error depends on observed adult BMI. This, therefore, has the potential to affect the results obtained for adult body size and breast cancer risk. However, the simulation results also suggest that misclassification of childhood size only masks the adult effect when both exposures have an effect on outcome in the same direction. Therefore, it was concluded that it is unlikely that measurement error is hiding a risk-increasing effect of adult body size on breast cancer, although it is possible that it is masking a larger protective effect [1].

We have added a brief discussion of potential limitations to using UK Biobank childhood body size data on lines 426-432:

“Childhood body size data from UK Biobank is based on a questionnaire completed by adult participants and could be subject to misclassification due to individuals misremembering their relative body size, which could potentially pose a great limitation to using this data. However, the genetic variants for childhood body size (originally identified by Richardson et al 2020 [16]) were validated in several different cohorts [63], [64] to be able to reliably separate childhood and adult body size, and also were robust to differential measurement error in simulations performed in the original study. Collectively, these analyses confirm the genetic variants from this data are suitable to be used to represent childhood body size.”

[1] T. G. Richardson, et al, ‘Use of genetic variation to separate the effects of early and later life adiposity on disease risk: Mendelian randomisation study’, *The BMJ*, vol. 369, 2020
Supplementary Data: <https://www.bmj.com/content/bmj/suppl/2020/05/06/bmj.m1203.DC1/rict052821.www.pdf>

[2] Bradfield JP et al, Early Growth Genetics Consortium. A genome-wide association meta-analysis identifies new childhood obesity loci. *Nat Genet* 2012;44:526-31.

[3] Speliotes EK, et al, MAGIC, Procardis Consortium. Association analyses of 249,796 individuals reveal 18 new loci associated with body mass index. *Nat Genet* 2010;42:937-48.

[4] M. Brandkvist et al., ‘Separating the genetics of childhood and adult obesity: a validation study of genetic scores for body mass index in adolescence and adulthood in the HUNT Study’, *Human Molecular Genetics*, vol. 29, no. 24, pp. 3966–3973, Dec. 2020,

[5] T. G. Richardson et al., ‘Evaluating the direct effects of childhood adiposity on adult systemic metabolism: a multivariable Mendelian randomization analysis’, *International Journal of Epidemiology*, Mar. 2021

[6] D. Bann et al, “Polygenic and socioeconomic risk for high body mass index: 69 years of follow-up across life“, *medRxiv*, November 2021, <https://www.medrxiv.org/content/10.1101/2021.11.08.21265748v1.full.pdf>

4. Why was the MR-egger intercept test used to quantify potential horizontal pleiotropy bias. We already know it performs worse than the Q-statistic (which was also used). Clearly saying that the Q-statistic does not constitute horizontal pleiotropy bias when the intercept test is close to zero, this should account for the intercept tests variability. Hence you would be back to simply using the intercept test. This seems fairly circular in reasoning. Also should we start the discussion on “absence of evidence does not provide proof of absence”. I understand the limitations of the Q-test, in this setting, but this does not seem to be doing much to address this, other than perhaps making the authors feel nice about themselves.

Response:

The Egger intercept and Cochran’s Q-statistic are both used as tests for pleiotropy within two-sample MR analyses because they are based on different assumptions and test different aspects of the data. We have therefore applied both in this setting to more thoroughly explore the potential for pleiotropy. A large Q-statistic provides evidence of heterogeneity in the per-SNP causal effect estimates, which could be due to either balanced or directional pleiotropy. Under the InSIDE assumption, a significant Egger intercept provides evidence of a directional pleiotropic effect. We agree however that these results should not be combined, as a large Q-statistic in the absence of a significant Egger intercept is not sufficient evidence that any pleiotropic effects are balanced. The results should rather be considered separately as two pieces of evidence taken as part of a set of sensitivity analysis to assess the robustness of the results. We have updated the text to reflect this (lines 297-298 and line 651).

5. This statement

“Lastly, although it is sometimes advised against calculating an indirect effect if the outcome is a binary variable (i.e. disease status) due to non-collapsibility of odds ratios [60], it has been shown that if the outcome effects have been quantified as log-odds ratios, it is acceptable to use betas of such outcome in the mediation analysis [68].”

This is not necessarily false, but it does not very much reflect the literature and even in the paper (despite the current authors being co-authors here). Clearly calculations of indirect effect will be biased when using odds ratios (or log odds ratio – do we really believe that taking a log addresses this at all? the OR is primarily defined on the log scale, so it is only natural to use this scale for any manipulation). What the reference paper merely shows – through LIMITED simulations is that the bias may be minimal, especially when the disease is sufficiently rare (they arbitrarily say, as many would, that 10% is sufficient).

Instead of this somewhat cavalier and annoying statement,

- 1) acknowledge that these analyses are biased,
- 2) discuss that the bias may be minimal (refer to the incidence in the source GWAS please), especially due to other source of bias,
- 3) perhaps discuss the potential for effect direction flipping due to under/over-estimation related to non-collapsibility.

If need I will dig-up multiple reference that actually support this, but (I am already late with the review) reference 68 actually already support this and the authors merely use it in an attempted bluff.

At the same time, provided the above is discussed, I do appreciate the actual conducted work and do not mind a likely modicum of bias.

Response:

Thank you for this comment, we acknowledge that the text originally played down this potential source of bias. We have adjusted the text in the mediation analysis methods section (starts on line 715) to acknowledge this source of bias in the analysis.

“Lastly, although it is difficult to perform mediation analysis on binary outcomes (i.e. disease status) due to the non-collapsibility of odds ratios [84], it has been shown that if the outcome effects have been quantified as log-odds ratios, it is acceptable to use them for estimating the indirect effects [64]. However, it is important to note that the analysis on log-odds ratios from both MVMR and two-step MR is likely to have some bias for both rare and common binary outcomes [64]. When the outcome is common, like in the present study (53.7% cases), it is expected that the estimates from Product and Difference methods are not going to align, and both are likely to be biased, with Difference method producing a conservative estimate [85].

Unsurprisingly, we observed a disagreement between the estimates from the two methods in the results and therefore performed simulation analysis to help us choose the method that produces more reliable results in our study. The mediation results and simulation analysis are described in Supplementary Materials Section E. The mediation analysis approach that was selected via simulation was the Product method with Delta method for the indirect effect SE/CI estimation.”

6. The figures are very nicely done, and appropriately scaled. Could the OR forestplot x-axis using a log-scale, with the tick labels exponentiated – this ensure the WALD-95%CI are symmetrical, preventing erroneous conclusions on a skewed in precision.

Response:

Thank you for this useful suggestion, the log-scale figure looks better (Figure 4, step2).

7. Figure 4, step 1 – I presume the effect size has a unit, could this be added? Same for Figure 6 – beta is not a unit, it is a letter in the Greek alphabet.

Response:

Thank you, this has been corrected as well (Figure 4 - step1 in question 6 and Figure 6 below).

Reviewer #3:

This is an interesting paper that adds to the literature on the effect of childhood adiposity on breast cancer, finding that effects of adiposity remain even when potential mediators are accounted for. The paper presents a framework for genetic epidemiology analyses that can also be used for addressing future similar research questions.

The paper is well written and presents comprehensive analyses and supporting data.

I suggest some specific comments for additional information to add, or points to clarify:

- ⇒ Thank you for the positive appraisal of the manuscript, we appreciate the valuable feedback. We provide the responses to your queries below and refer to incorporated changes from your suggestions by line numbers.

1. Was the childhood adiposity instrument based on the variants identified in females only?

Response:

This is correct. We have made it clearer in the manuscript (line 513).

“The GWAS of childhood body size and adult body size used in this study were performed by Richardson et al (2020) [16] on UK Biobank data (N=246,511; female-only data, including for instruments extraction).”

2. Regarding GWAS generated in UK Biobank (e.g. sex hormones, IGF-1, number of births etc) that were used to create genetic instruments, what checks were done to ensure the validity of these signals and were they consistent with published analyses? For example, published GWAS of sex hormone measures in UKB have included various additional covariates (e.g. fasting time, menopause status).

Response:

This is a great point, and we acknowledge that making sure that the signals between the GWAS we generated and the published GWAS are comparable is very important. To address this, we performed LDSC regression (Bulik-Sullivan *et al* 2015) using the data from Ruth *et al* 2020 (total and bioavailable testosterone, SHBG) and Murphy *et al* 2019 (IGF-1 – data from Neale Lab www.nealelab.is/uk-biobank) studies for comparison:

	rg	lo_ci	up_ci	se	pval
IGF1	1.00	0.99	1.00	0.0025	< 1e-300
SHBG	1.00	0.99	1.01	0.0046	< 1e-300
Bioavailable testosterone	1.00	0.99	1.01	0.0031	< 1e-300
Total testosterone	0.99	0.98	1.01	0.0064	< 1e-300

The *rg* score between our study data and the published data is very close to 1, i.e. the datasets are ideally correlated, despite slight differences in sample size and covariates.

This analysis can be included as a supplementary table in the manuscript at the reviewer/editor's discretion.

3. Given that puberty timing is likely to be related to many of the other mediators investigated, e.g. IGF-1, hormone levels, have the authors considered exploring this in MVMR?

Response:

This is a valid point, as puberty timing (i.e. ‘age at menarche’) may quite likely be related to many of the considered mediators. However, within this study’s design, we chose not to pursue investigating this for the following reasons:

- 1) Adding 'age at menarche' as a covariate / extra exposure in MVMR analyses with other mediators may not work very well, as we would likely lack power in the analysis and have problems with instrument strength. Generally, the more exposures are used in the analysis, the harder it is to get reliable estimates in MVMR.
- 2) 'Age at menarche' has an unusual mediating relationship in this context, as the indirect effect from it is drawn in the opposite direction from the direct effect of childhood adiposity on breast cancer risk (see question 5 for more detail on this). Therefore, it could complicate the picture when investigating other potential mediators of the effect of childhood adiposity and may hinder interpretation.
- 3) Puberty timing is not the main focus of this study, as we are primarily interested in childhood adiposity effects. We agree that it would be interesting to explore 'age at menarche' associations with hormones in future work.

4. Supp Fig S3 and lines 231-232: Does attenuation of the effect estimates for age at menarche and IGF-1 with the addition of childhood height in the MVMR analyses suggest a more complex relationship between age at menarche and size in early life with breast cancer?

Response:

The relationship between childhood height and age at menarche is complex as these traits are closely related. The height GWAS used in this study is based on categorical questionnaire data (described on lines 551-552), and therefore it has its limitations. The questionnaire relies on the adult participants' recollection of how tall they were at the age of 10 compared to their peers. Therefore, it may be a very biased measure, as taller kids reach puberty sooner, but also if you reach puberty sooner, you might remember being taller because you got taller younger. Therefore, the underlying data may be biasing these two traits to correlate further.

This makes it challenging to interpret the MVMR results of the analyses that include childhood height. Overall, the instrument strength seems adequate (Table S20, also shown below), although the Fst in the analysis involving 'age at menarche' is lower for all exposures than in the analyses with other mediators. This implies that there is substantial shared variance between childhood height and age at menarche, so conditionally their instrument strength is weaker, although this does not explain the effect attenuation observed.

We agree that these MVMR results suggest a potentially complex relationship between height, body size, and age at menarche with breast cancer. However, with the current data we cannot explore this relationship further than that shown in this analysis. The attenuation of the effect is interesting, but we do not want to overinterpret these results as there are quite wide CIs which overlap with estimates from the other analyses not including the height measure. Also, with the lower instrument strength in the trivariate MVMR analysis, we decided to not pursue exploring this analysis further.

We are open to removing this supplementary analysis from the paper at the reviewer/editor's discretion, if the underlying childhood height data is considered unreliable, or the weaker instruments raised concern in some MVMR analyses.

mediator	Childhood body size (Fst)	Childhood height (Fst)	mediator	Qstat	Qpval
IGF1	12.70	75.94	33.98	1462.59	7.17E-108
SHBG	13.09	76.69	31.86	1252.72	1.72E-77
Testosterone (bioavailable)	13.48	80.61	13.01	1253.44	2.09E-79
Testosterone (free)	13.40	80.96	13.41	1524.99	1.27E-120
Testosterone (total)	13.24	81.30	12.62	1493.29	1.91E-115
Age at menarche (Perry)	10.06	52.38	10.07	1163.32	3.22E-97
Age at menopause (Day)	14.81	84.61	6.19	1128.66	1.99E-93

5. Lines 277-280, 304-305: How do you interpret the breast cancer increasing indirect effect of childhood size via age at menarche?

Response:

The MR estimates showed that higher childhood body size is associated with a lower age at menarche (effect size per standard deviation, -0.79, 95%CI [-0.95: -0.64]), while higher age at menarche is associated with decreased breast cancer risk (effect size per standard deviation, -0.08 [-0.16: -0.01]). When we estimate the indirect effect of childhood body size via age at menarche, we multiply two negative coefficients and get a positive indirect effect value (effect size per standard deviation, 0.06 [0.01: 0.13]).

The biological interpretation of this is that higher childhood body size lowers the age at menarche, while starting menarche at a younger age is associated with a higher risk of breast cancer.

6. In the split sample analyses, did the instruments use variants identified at $P < 5 \times 10^{-8}$ from the full sample analysis or from the split sample analysis?

Response:

In split sample analyses, we used variants identified at $P < 5 \times 10^{-8}$ from the split sample exposure GWAS. This has been made clearer on line 694:

“The exposure instruments were extracted from the new split-sample exposure GWAS ($p < 5 \times 10^{-8}$), and step 1 of two-step MR and MVMR were repeated, [...]”

7. Line 227: I couldn't see 'age at first birth' in Table S11 or S12 and this trait isn't listed as prioritised in Supp Table S5.

Response:

Thank you for spotting this and we apologise for the confusion. The mention of 'age at first birth' has now been removed from the text, as it was not a prioritised mediator. Modified lines: 270-273.

8. Lines 231-232: I couldn't find the childhood height measures in the methods.

Response:

Thank you for bringing this to our attention. This has now been included in the *Methods/Description of selected traits* section on lines 551-552:

“Childhood height is another categorical trait from a questionnaire completed by adult UK Biobank participants, with three categories describing comparative height at age 10 ('shorter', 'about average', 'taller').”

Reviewers' comments:

Reviewer #1 (Remarks to the Author):

This paper used two-sample MR, two-step MR and multivariable MR to analyze how early life adiposity influences breast cancer risk. And found that most of the hypothesised mediators are affected by childhood adiposity, only IGF-1, testosterone, age at menarche and age at menopause influence breast cancer risk. This study gives some clues to the breast cancer risk. However, there are some limitations:

1. In abstract, some statistical results should be written in the corresponding factors. It is hard to understand why write this sentence "This suggests that, individually, none of the investigated traits strongly mediate the effect. It is plausible that several related traits collectively contribute to the effect, or a pathway not considered in this study is involved. I suggest to rewrite the abstract.
2. It is hard to understand which mediator traits the author included. In introduction, denote "sex hormones, reproductive traits, glycaemic traits, and physical traits", but in study overview of results is different.
3. It is not needed to give the figure 2 and 3, the method is generally accepted.

Reviewer #2 (Remarks to the Author):

The authors have more than adequately addressed all my comments. Really lovely contribution. I do apologise for my delayed review - blaming annual leave.

Reviewer #3 (Remarks to the Author):

The authors have thoroughly answered my questions and I thank them for the extra analyses presented. The revisions made are appropriate.

Response to Reviewer #3 point 2: The genetic correlations with the published hormone GWAS are convincing – to avoid adding another Supp Table, it might be good to note somewhere that this validation was carried out (if the editor agrees)?

I have no further comments to make.

Response to reviewers

Please find below our detailed responses to all Reviewers' comments and suggestions marked in blue. *Italicised* text indicates text quoted from the manuscript. The line numbers mentioned in responses refer to the revised text. All changes in the revised text are highlighted.

Reviewer #1:

This paper used two-sample MR, two-step MR and multivariable MR to analyze the how early life adiposity influences breast cancer risk. And found that most of the hypothesised mediators are affected by childhood adiposity, only IGF-1, testosterone, age at menarche and age at menopause influence breast cancer risk. This study give some clues to the breast cancer risk. However, there are some limitation:

⇒ Thank you for the positive appraisal of the manuscript, we appreciate your feedback.

1. In abstract, some statistic results should write in the corresponding factors. It is hard to understand why write this sentence "This suggests that, individually, none of the investigated traits strongly mediate the effect. It is plausible that several related traits collectively contribute to the effect, or a pathway not considered in this study is involved. I suggest to rewrite the abstract.

While the abstract word limit does not allow us to include the results for all traits investigated in this study, we have now revised this to include the estimate values for the key traits previously specified. We apologise for the lack of clarity in this sentence. We have now revised this to make it more explicit, and hope that the rephrased sentence is easier to follow.

2. It is hard to understand which mediator traits the author included. In introduction, denote "sex hormones, reproductive traits, glycaemic traits, and physical traits", but in study overview of results is different.

We appreciate this comment and acknowledge it may not be clear for the first-time readers to follow which mediators across the four categories were investigated. To address this, we now refer to Figure 1 both in Introduction and the study overview section in Results. Modified lines: 120, 130

3. It is not needed to give the figure 2 and 3, the method is generally accepted.

We believe Figures 2 and 3 provide a useful visual aid to the readers - particularly to those less familiar with MR approaches – which allows them to quickly grasp our study design and the methods that we used. Nevertheless, we have grouped them into a single figure (now Figure 2) so that all schematics are presented at once for improved flow and clarity.

Reviewer #2:

The authors have more than adequately addressed all my comments. Really lovely contribution. I do apologise for my delayed review - blaming annual leave.

⇒ Thank you for the positive appraisal of the manuscript, we appreciate the valuable feedback that has helped us strengthen our work.

Reviewer #3:

The authors have thoroughly answered my questions and I thank them for the extra analyses presented. The revisions made are appropriate.

Response to Reviewer #3 point 2: The genetic correlations with the published hormone GWAS are convincing – to avoid adding another Supp Table, it might be good to note somewhere that this validation was carried out (if the editor agrees)?

I have no further comments to make.

⇒ Thank you for the positive appraisal of the manuscript. We have added the genetic correlation analysis as a separate section in Supplementary Materials (Section F) and refer to it in the main manuscript (lines 526-528).